# Focal cortical seizures start as standing waves and propagate respecting homotopic connectivity

L. Federico Rossi [1], Robert C. Wykes[2], Dimitri M. Kullmann[2] & Matteo Carandini [1]

Focal epilepsy involves excessive cortical activity that propagates both locally and distally. Does this propagation follow the same routes as normal cortical activity? We pharmacologically induced focal seizures in primary visual cortex (V1) of awake mice, and compared their propagation to the retinotopic organization of V1 and higher visual areas. We used simultaneous local field potential recordings and widefield imaging of a genetically encoded calcium indicator to measure prolonged seizures (ictal events) and brief interictal events. Both types of event are orders of magnitude larger than normal visual responses, and both start as standing waves: synchronous elevated activity in the V1 focus and in homotopic locations in higher areas, i.e. locations with matching retinotopic preference. Following this common beginning, however, seizures persist and propagate both locally and into homotopic distal regions, and eventually invade all of visual cortex and beyond. We conclude that seizure initiation resembles the initiation of interictal events, and seizure propagation respects the connectivity underlying normal visual processing.

[1] UCL Institute of Ophthalmology, University College London, 11-43 Bath Street, London EC1V 9EL, UK. [2] UCL Institute of Neurology, University College London, Queen Square, London WC1N 3BG, UK. Correspondence and requests for materials should be addressed to L.F.R. (email: luigi.rossi.12@ucl.ac.uk) or to R.C.W. (email: r.wykes@ucl.ac.uk)

Focal cortical epilepsy, also known as partial-onset epilepsy, frequently resists pharmacological treatment. Its origins are thought to be unambiguously cortical: seizures arise from congenital or acquired cortical lesions such as focal cortical dysplasia, penetrating brain injuries, abscesses, strokes, and tumors[1]. Much of its morbidity results from the spread of seizure activity from a cortical focus to further cortical regions. These regions can be local and contiguous to the focus, as in the "Jacksonian march" seen in motor cortex. However, they can also be distal and ultimately involve both hemispheres, and subcortical centers ("secondary generalization"), causing loss of consciousness[2].

It is not known whether seizures spread along the same circuits that support information processing during normal cortical activity[3]. These circuits may not act as effective constraints during

seizures. First, the circuits that underlie normal function rely on synaptic transmission, whereas seizures may involve non-synaptic mechanisms[4–11]. Second, the flow of activity along the circuits that underlie normal function depends on the balance of synaptic excitation and inhibition. Because seizures disrupt this balance[12–16], their propagation may obey different rules.

A related question concerns the difference between seizures (ictal events), and the numerous, brief interictal events that characteristically occur between seizures. Seizures represent excessive neuronal firing, lasting seconds or minutes and spreading locally and distally[12, 17]. Interictal events, instead, are brief and localized[18]. A longstanding theory is that their duration and spread are limited by a powerful inhibitory surround[19]. It is not known, however, if seizures and interictal events originate in similar neuronal populations, and if their initiation differs in

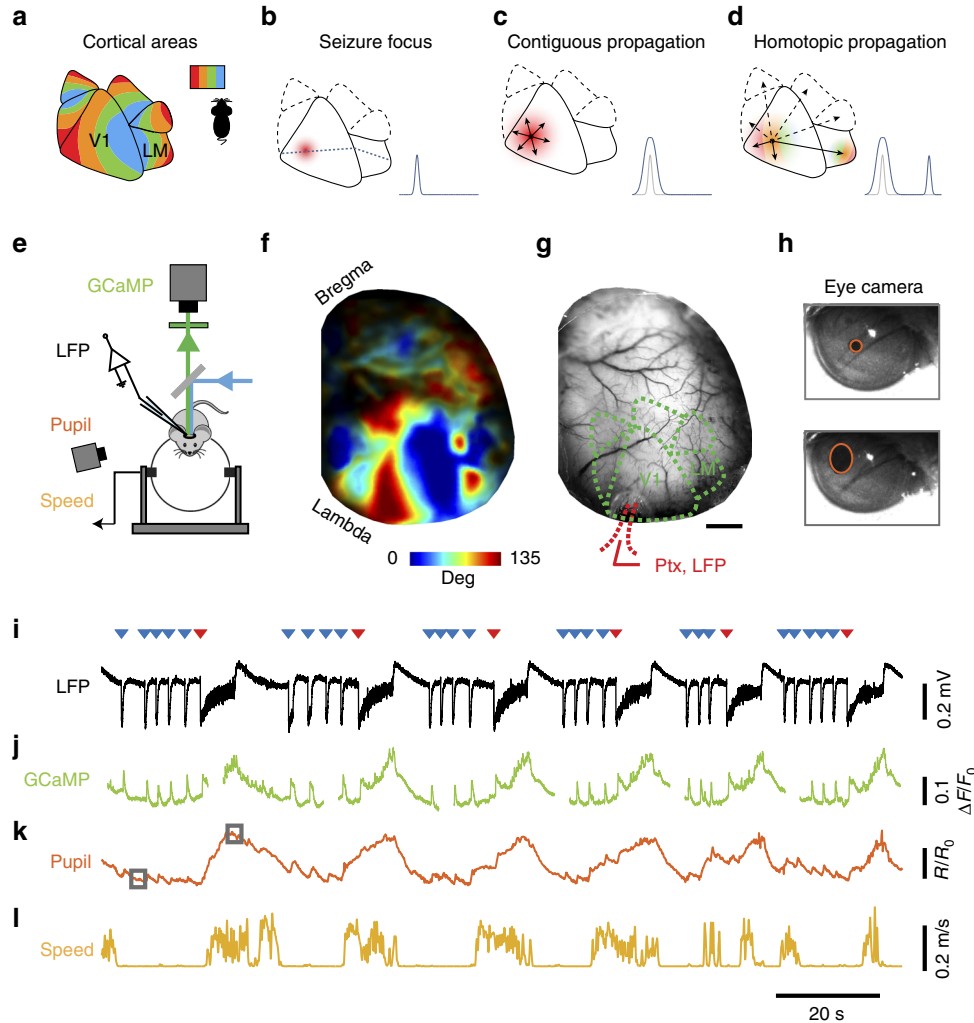

**Fig. 1** Testing hypotheses for cortical seizure propagation with simultaneous imaging, recordings and behavioral measurements in the awake mouse. **a** Cartoon of mouse visual cortex showing six visual areas in the right hemisphere, each containing a map of the left visual field. *Colors* indicate territories that prefer the same horizontal position. V1 and LM are the primary and secondary visual areas. **b** Cartoon depicting elevated activity in the seizure focus in V1. *Dotted line* through the epileptic focus connects V1 and LM regions that prefer the same vertical position. *Inset*: Profile of activity along that line. **c** Contiguous propagation hypothesis: local mechanisms make seizure spread radially to nearby territories. *Inset*: profile of idealized activity at onset (*gray*) and during spread (*blue*). **d** Homotopic propagation hypothesis: long-range connections generate secondary distal foci in homotopic locations in higher visual areas. *Inset as in* **c**. **e** Schematic of the experimental set-up. **f** Example retinotopic map obtained from a GCaMP6f mouse. *Colors* indicate preferred horizontal position (*color bar*). **g** Image acquired during simultaneous widefield imaging and LFP recordings. *Red dots* highlight location in V1 of pipette for injection of Picrotoxin (Ptx) and recordings of local field potential (LFP). Scale bar indicates 1 mm. **h** Frames from the eye camera taken when the pupil was constricted (*top*) or dilated (*bottom*), with fitted ellipses (*red*). **i** LFP traces measured during epileptiform activity, showing seizures (*red triangles*) and interictal events (*blue triangles*). **j** Simultaneous GCaMP fluorescence, averaged over the imaging window (discontinuities indicate pauses in image acquisition). **k,l** Behavioral measures during epileptiform discharges: pupil dilations (**k**) and running speed (**l**). *Squares* in **k** indicate times of example frames in **h**

terms of spatial profile and temporal evolution[20]. More generally, it is not clear why seizures persist and propagate, while interictal events are brief and stay local.

We addressed these questions in the visual cortex of the awake mouse, where we could readily distinguish between two scenarios of seizure propagation (Fig. 1a–d). Mouse visual cortex includes multiple retinotopic areas[21, 22], the largest of which are the primary and latero-medial areas (V1 and LM, Fig. 1a). Epileptiform activity in a focal region of V1 (Fig. 1b) could then spread in at least two ways: contiguously or homotopically. In the first scenario, epileptiform activity propagates through local mechanisms to V1 regions that are contiguous to the focus (Fig. 1c). In the second scenario, epileptiform activity propagates along long-range projections from V1 to homotopic regions of LM and higher visual areas, i.e. regions that have the same retinotopic preference[21, 23], producing one or more distinct, secondary foci (Fig. 1d).

To evaluate these models of seizure propagation, we induced epileptiform activity in a focal region of V1 and used widefield imaging of genetically encoded calcium indicators[24, 25] to characterize activity in space and time. We found that prolonged seizures and brief interictal events both start as standing waves in the V1 focus and in homotopic locations in higher areas. Following this common beginning, however, seizures persist and propagate both locally and into homotopic distal regions. Seizures, moreover, contain prominent oscillations, which also propagate along homotopic connectivity. We conclude that seizure initiation resembles the initiation of interictal events, and seizure propagation respects the connectivity underlying normal visual processing.

## Results

**Measuring connectivity and epileptiform activity.** We generated transgenic mice expressing a genetically encoded calcium indicator (GCaMP3 or GCaMP6f) in excitatory neurons of the cortex[26–28], and implanted a head-post and a glass cranial window spanning the left parietal bone, which we thinned to facilitate optical access[25]. After recovery and a period of habituation, we drilled a small opening in the glass to insert of a micropipette, and head-fixed the mice over a treadmill where they were free to run while we performed visual stimulation, widefield imaging, and electrophysiological recordings (Fig. 1e).

We first used widefield imaging of GCaMP fluorescence to measure activity evoked by visual stimulation, and mapped the retinotopic organization of V1 and higher visual areas (Fig. 1f). For each mouse, we aligned the salient features of the retinotopic map[25] on a reference map of the areal organization of the mouse visual cortex[22]. This procedure identified multiple visual areas (Fig. 1f) including V1 and adjacent area LM, which is thought[29] to be homologous to primate area V2. The retinotopic map gauges the connectivity between V1 and higher visual areas: regions that respond to the same retinotopic location are also interconnected[21, 23].

We then elicited focal epileptiform discharges by blocking GABA_A receptors in a small region of V1 (Fig. 1g). Local application of GABA_A receptor antagonists[30] and blockers[31] is well-known to cause focal epileptiform events. We delivered a blocker, picrotoxin (10 mM) in a pipette inserted into medial V1 through the hole in the glass window, and we applied no pressure, to minimize diffusion beyond the tip of the pipette (resistance 0.5–3 MΩ). We targeted layer 5 because its extensive recurrent, horizontal and columnar connectivity enable it to evoke prolonged depolarization across layers[32], making it a promising location for triggering epileptiform events.

Within minutes of pipette insertion, recordings of local field potential (LFP) made from the same pipette revealed robust

epileptiform events (Fig. 1i). These events consisted of prolonged seizures (Fig. 1i, *red triangles*) separated by sequences of brief interictal discharges (Fig. 1i, *blue triangles*). These epileptiform events were mirrored by large amplitude GCaMP optical signals (Fig. 1j). They were typically accompanied by pronounced increases in pupil dilation (Fig. 1k) and by bouts of running (Fig. 1l).

**Signatures of seizures and interictal events.** These epileptiform events could be readily categorized into distinct classes: (Fig. 2). Their duration was distributed bimodally, with a median duration of 8.6 s for seizures, and 0.5 s for interictal events (Fig. 2a, $p < 0.001$, Wilcoxon rank sum test, $n = 5$ mice). Interictal events occurred on average $13.4 \pm 3.7$ times per minute (mean $\pm$ SEM, $n = 5$ recordings in 5 animals), much more frequently than seizures, which occurred once per minute ($1.0 \pm 0.2$ events min$^{-1}$). Interictal events tended to be slightly larger than seizures (Fig. 2b, $p < 0.001$, Wilcoxon rank sum test, $n = 5$ mice) and were followed by briefer periods of relative LFP quiescence (Fig. 2c).

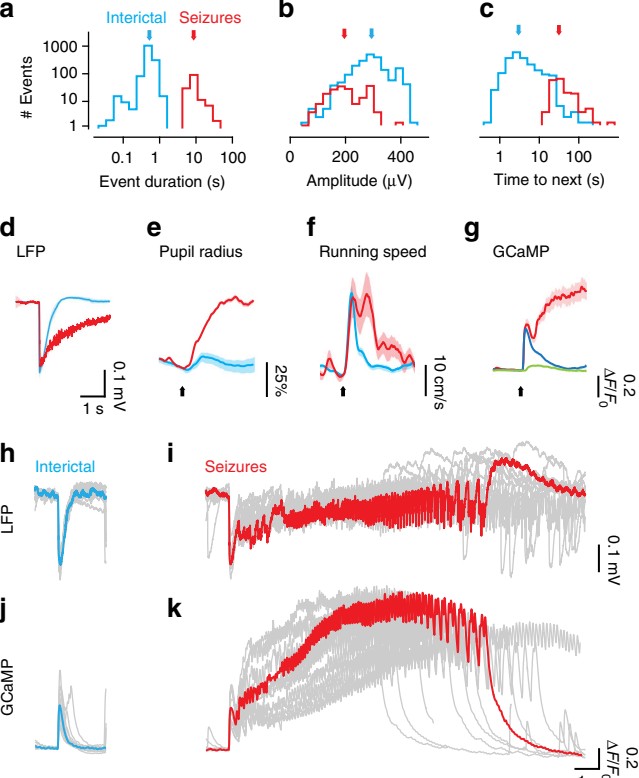

**Fig. 2** Neural and behavioral signatures of seizures and interictal events. **a** Distribution of the duration of interictal events (*blue*) and seizures (*red*). *Arrows* indicate the medians of each distribution. **b** Distribution of event amplitudes, measured as the peak of the initial LFP negative event. **c** Distribution of inter-event intervals (time to next event). **d** LFP waveform following the onset of interictal events (*blue*) and seizures (*red*), averaged across all events in five mice. **e** Change in pupil radius triggered on the onset of interictal events (*blue*) and seizures (*red*), for one representative animal. *Arrow* indicates event onset. **f** Same, for running speed. **g** Time courses of GCaMP activity in the same animal, averaged over area V1, during interictal events (*blue*) and seizures (*red*). *Green trace* shows response to visual stimuli for comparison. **h** LFP waveform of representative interictal events in one experiment. *Blue trace* highlights a single event. **i** Same as **h**, for seizures in the same experiment, with a highlighted trace in *red*. **j**, **k** GCaMP activity averaged over visual cortex during the example events in **h**

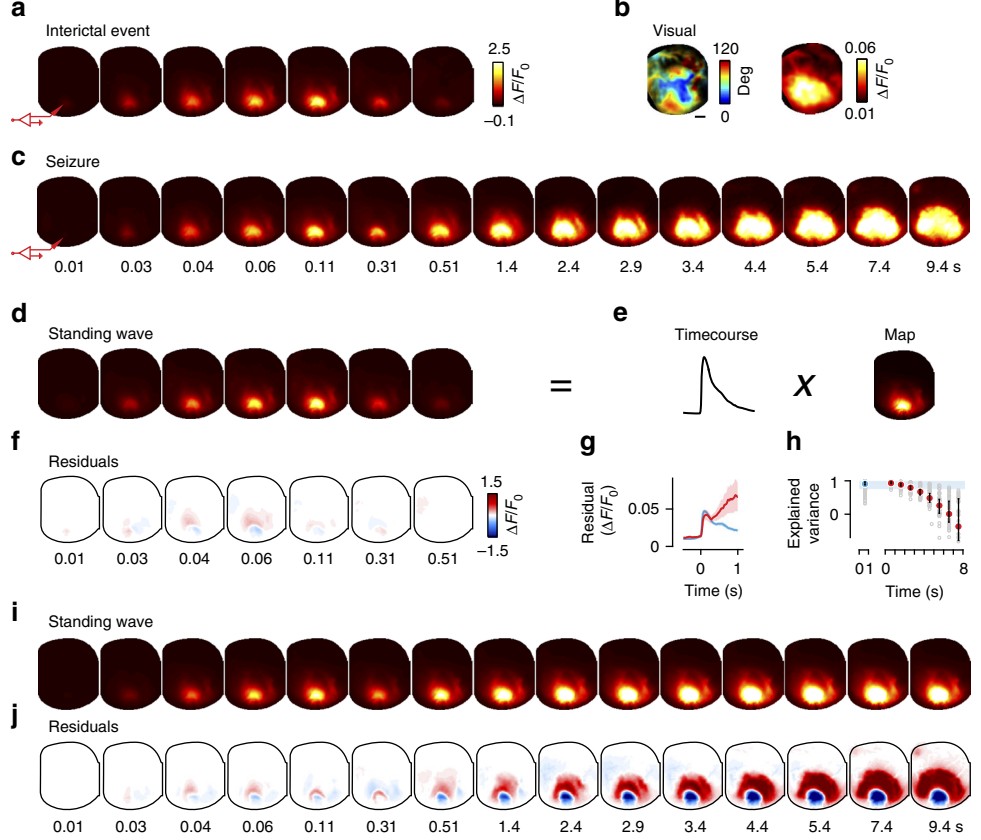

**Fig. 3** Interictal events and seizures start as standing waves, and seizures subsequently propagate widely across cortex. **a** Frames obtained through GCaMP imaging in a representative interictal event (the one with LFP in Fig. 2h and j). The *cartoon electrode* indicates the site of Ptx injection and epileptic focus. **b** Retinotopic map and map of maximal activation in response to visual stimulation for this animal. *Scale bar* is 1 mm. **c** Same as **a**, for a representative seizure (the one highlighted in Fig. 2i and k). *Labels* indicate the time of each frame from event onset and apply to frames in **a** and **c**. **d** Predictions of a standing wave model fit to interictal event in **a**. **e** The standing wave is the product of a single temporal waveform (Time course) and a fixed spatial profile (Map). The map and the time course shown are averaged across interictal events for this animal. **f** The residuals of the fit in **e** are small, indicating little deviation of interictal event from standing wave. **g** The root-mean-square residuals for the standing wave model applied to interictal events (*blue*) and seizures (*red*). The spatial map was optimized to fit interictal events. *Shaded areas* indicate two s.e.m. **h** Variance explained by the standing wave model for interictal events (*blue dot*) and 1 s intervals of seizures (*red dots*). *Error bars* show median ± 1 quartile. *Shaded blue area* indicates the 96% confidence interval for quality of the fit to interictal events. **i** Predictions of the standing wave model for the seizure in **c**. The model was constrained to have the same spatial map as interictal events, and was free to have the best-fitting time course. **j** Residuals between seizure and standing wave model are small in the first ~0.3 s after onset, but subsequently become prominent, when the standing wave model fails to capture the propagation typical of seizures

Seizures had pronounced behavioral correlates: they were typically accompanied by prolonged pupil dilations (Fig. 2e, Supplementary Fig. 1a) and by bouts of running (Fig. 2f, Supplementary Fig. 1b, Kolmogorov–Smirnov test, $p < 0.01$, $n = 5$ mice). In addition, the longer seizures culminated in abnormal behaviors such as tail flicking and clonic movements of the forelimbs. For interictal events, pupil dilations were typically smaller (Fig. 2e), running bouts briefer (Fig. 2f) and not consistent across mice (Supplementary Fig. 1b).

Interictal events and seizures also had distinct temporal and spectral features (Fig. 2h and i, Supplementary Fig. 2). The LFP waveform of interictal events was highly stereotypical: a sharp negative deflection lasting 50–100 ms (Fig. 2h, Supplementary Fig. 2a). Seizures had similar onset (Fig. 2i, Supplementary Fig. 2b), but this onset was followed by a rapid escalation of high-frequency activity, with a large increase in power between 6 and 30 Hz (Supplementary Fig. 2b,e). The increase in power near 10 Hz was consistent across events and animals (Supplementary Fig. 2i).

These epileptiform events were faithfully tracked by large fluorescence signals, which dwarfed those seen during normal visual processing (Fig. 2g). The peak fluorescence changes ($\Delta F/F_0$) seen during epileptiform events were $125 \pm 22\%$ for interictal events and $175 \pm 50\%$ for seizures. These signals were much larger than those evoked by visual stimuli[25, 33] (Fig. 2g), which peaked at $4 \pm 1\%$. Yet, they a bore close relationship to the LFP (Supplementary Fig. 3b), and faithfully followed the main frequency components of seizures (Fig. 2j and k).

The basic features of these epileptiform events were not peculiar to the acute disinhibition induced by picrotoxin: they were also observed in additional experiments where we elicited focal epileptiform activity with injections of pilocarpine[34], an agonist of muscarinic receptors (Supplementary Fig. 4). In these experiments, epileptiform events started as brief interictal events (Supplementary Fig. 4a), which became progressively longer to resemble seizures measured with picrotoxin (Supplementary Fig. 4b). The LFP spectral signatures of these events closely resembled those seen with picrotoxin (Supplementary Fig. 4d–f). As with picrotoxin, moreover, widefield changes of fluorescence associated with these events dwarfed the responses measured with visual stimuli. However, they were about half the size of those measured with picrotoxin, and the high density of the pilocarpine

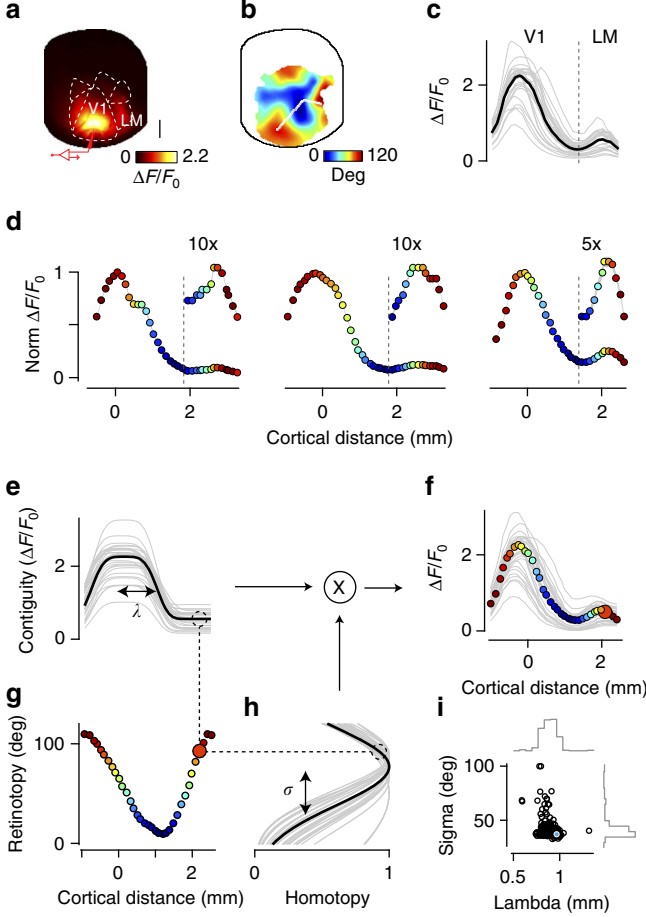

**Fig. 4** Interictal events and early seizures engage both contiguous spread and homotopic connectivity. **a** Average spatial profile of interictal events measured in one mouse, obtained by averaging the maximum projection map across events (*n* = 242). The *cartoon electrode* indicates the site of Ptx injection and LFP recording. *Scale bar* is 1 mm. **b** Map of retinotopy (preferred horizontal position) with a line from the focus in V1 to area LM, joining regions of interest (ROIs) with same preferred vertical position as the focus in V1. **c** Peak $\Delta F/F_0$ response at each ROI in **b** for a representative interictal event (*black*) and for 30 other interictal events (*gray*). **d** The spatial profile averaged across interictal events for three representative animals (two measured with GCaMP3 and the third with GCaMP6), along lines drawn with the same strategy as in **b**. *Dot color* indicates the retinotopic preference of the corresponding location. *Insets* show the magnified profile of responses in area LM. The very thin *gray shaded line* represents mean ± s.e.m. **e–h** The bimodal profile of activity for each interictal event is well described by the dot product of a function of cortical distance from the focus **e**, and a function of retinotopic distance from the focus **h**. *Dotted lines* illustrate how the model predicts the second peak of activation ~2 mm away from the focus **f**, due to the similar retinotopic preference of that region and the focus **g**. **i** Distribution of fit parameters for the example animal, indicating a consistent role of contiguous spread and homotopic connectivity across events

solution made it impossible to inject it through the same pipette used to record the LFP. For these reasons, in the following, we report on results obtained with picrotoxin.

**Interictal events and seizures start as standing waves**. The fluorescence signals allowed us to study the spatial spread of epileptiform events, and revealed profound differences between seizures and interictal events (Fig. 3a–c). The transient fluorescent

signals seen during interictal events were localized in V1 with a smaller peak in LM (Fig. 3a). They involved a region that was markedly smaller than the entirety of visual cortex (Fig. 3b, Supplementary Movie 1). Seizures, instead, progressively invaded the entire visual cortex and beyond (Fig. 3c, Supplementary Movie 2). Seizures that lasted > 30 s, in particular generalized to most (or all) of the imaged hemisphere (Supplementary Fig. 5). Seizures and interictal events had similar origins: 94 ± 4% of events started in V1, close to the pipette used to apply picrotoxin and record the LFP (starting positions clustered within a radius of 0.32 ± 0.12 mm).

For interictal events, the evolution of activity in space and time was a simple standing wave: (Fig. 3d–h). In a standing wave, the responses in all locations follow a similar time course, so that the sequence of images is the product of a single spatial profile (an image) and a single time course (Fig. 3d and e). We fitted this model to the interictal events and obtained predictions that resembled closely the actual measurements (Fig. 3d), accounting for 94.0 ± 0.4% of the variance of the data (Fig. 3h, *blue dot*). Indeed, the residuals between the data and the predictions of the model were small (Fig. 3f), with the maximal residual of $\Delta F/F <$ 10% (Fig. 3g, *blue trace*), a negligible fraction of the typical amplitude ($\Delta F/F$ ~125%) of interictal events.

Seizures initially closely resembled interictal events, but then propagated extensively (Fig. 3g–i). A standing wave model with the spatial profile of interictal events was appropriate to describe the beginning of seizures, but it became inadequate after a few hundred milliseconds (Fig. 3i and j): it progressively overestimated the activity at the focus (negative residual, Fig. 3j), while underestimating the activity in the surrounding region (positive residual, Fig. 3j). Consequently, the quality of this model's predictions deteriorated sharply with time (Fig. 3g and h). Seizures, thus, start as standing waves just as interictal events, but then they propagate beyond a common initial spatial profile to engage wider regions of cortex.

**Interictal events engage contiguous and homotopic regions**. The spatial profile common to interictal events and early seizures typically encompassed not only the focus, but also one or more distinct regions that were homotopic, i.e. that shared the same retinotopic preferences (Fig. 4a–d). For instance, two distinct lobes are visible 60–100 ms into the example epileptiform events (Fig. 3a and c). A larger lobe lies in area V1 and a smaller lobe in area LM (Fig. 4a). To characterize these two lobes, we drew a line connecting V1 to LM along the direction of maximal change in preferred horizontal position previously obtained when retinotopy was mapped (Fig. 4b). The profile of activation along this line was bimodal (Fig. 4c), with the activation in V1 being 8 ± 2 times higher and ~2 times wider than in LM (*n* = 4 mice, Supplementary Fig. 6a–d). Crucially, the two activations involved regions with matching retinotopic preference (Fig. 4d). The retinotopic preferences of the two peaks, indeed, were not significantly different (*p* = 0.4, Wilcoxon signed rank test), and were significantly correlated (*r* = 0.48, *p* < 0.001, Supplementary Fig. 6e,f).

These observations indicate that two factors determine the spatial profile of interictal events and early seizures: one local, causing contiguous spread and one distal, causing homotopic spread (Fig. 4e–i). The local factor could, in principle, arise from disinhibition (spread of picrotoxin, loss of the last A should be a subscript and not capitalized: GABA$_a$ receptor electrochemical driving force, or depolarization block of interneurons), from local excitatory circuits, or from non-synaptic mechanisms; its impact falls off with cortical distance (Fig. 4e). The distal factor, by contrast, arises from the same axonal and synaptic organization

that supports homotopic connectivity, whose impact decreases with retinotopic distance (Fig. 4h).

We combined these two effects into a simple multiplicative model (Fig. 4e–h), and found that with only six parameters the model provided good fits to the data (Fig. 4f). The function of retinotopy, in particular, was described by one key parameter, the standard deviation $\sigma$. As this parameter grows, the function becomes closer to a constant, and retinotopy plays a lesser role in the profile of the epileptiform events. When fitting the data, $\sigma$ consistently took values < 40° (Fig. 4i) confirming an important contribution of retinotopy (Fig. 4h), which in turn produced a clear secondary peak in activity (Fig. 4f). Homotopic connectivity thus plays a key role in determining the spatial profile of interictal events and early seizures. Nonetheless, at this early stage, homotopic connectivity is not sufficient to recruit in downstream territories the same levels of abnormal activity recorded at the focus. Retinotopic modulation was present in all mice, but the strength of its contribution varied across animals (Fig. 4d), possibly reflecting experimental variability in the induction of the focus and levels of expression or sensitivity of the calcium indicator.

**Seizures propagate along homotopic connectivity.** These same two factors—contiguity and homotopy—also determined the subsequent evolution of seizures, when the standing wave gave way to escalating propagation of activity (Fig. 5a–d). As we have seen, seizures propagated to engage most of the visual cortex, and often beyond, producing activity levels comparable to those observed at the focus (Fig. 5a and Supplementary Fig. 7). The pattern of propagation can be seen by following the activity of individual loci between the focus in area V1 and the homotopic location in area LM (Fig. 5c). In the example shown, the focus in V1 had a retinotopic preference ~100° away from the middle of the visual field, which is indicated in orange in the map of retinotopy (Fig. 4b). Accordingly, at that location the calcium traces show seizure activity with the earliest onset (Fig. 5c, *top arrow*). This activity progressively invades further V1 locations, eventually reaching the border with area LM, where receptive fields are in the middle of the visual field (0°, Fig. 5c, *blue traces*). Well before entering those areas, however, the seizure has already reached a distal region: the portion of LM with the same retinotopic preference as the focus (Fig. 5c, *bottom arrow*). For instance, 6–8 s into the event, the seizure activity shows a bimodal profile, being stronger in the homotopic regions of LM than in regions of V1 that are closer to the focus (Fig. 5d).

The two factors, contiguity and homotopy, both contributed approximately linearly to the time course of seizure propagation (Fig. 5e–j). To measure the time it took a seizure to invade different regions, we defined 'delay to seizure invasion' as the time taken for each pixel to reach 60% of its maximum activity. Consistent with the observations made above, a map of this quantity shows activations occurring earlier in the V1 focus and in the homotopic region of LM than in other locations (Fig. 5h), and a plot of delay along a line from V1 to LM shows the delay growing with distance while displaying two distinct troughs (Fig. 5e). When averaging these measures across mice, we found that delay strongly correlated with distance ($r = 0.81$, $p < 10^{-7}$, $n = 64$ seizures in four mice), with an average propagation speed of $0.48 \pm 0.03$ mm s$^{-1}$ (Fig. 5f). However, this linear prediction based on distance underestimates the delays of proximal territories and overestimates those of distal territories (Fig. 5f). The corresponding residuals were highly correlated with retinotopic distance from the focus ($r = 0.82$, $p < 10^{-24}$, Fig. 5g). Indeed, if delay to cortical invasion were solely due to cortical distance, it would grow radially from the focus and display

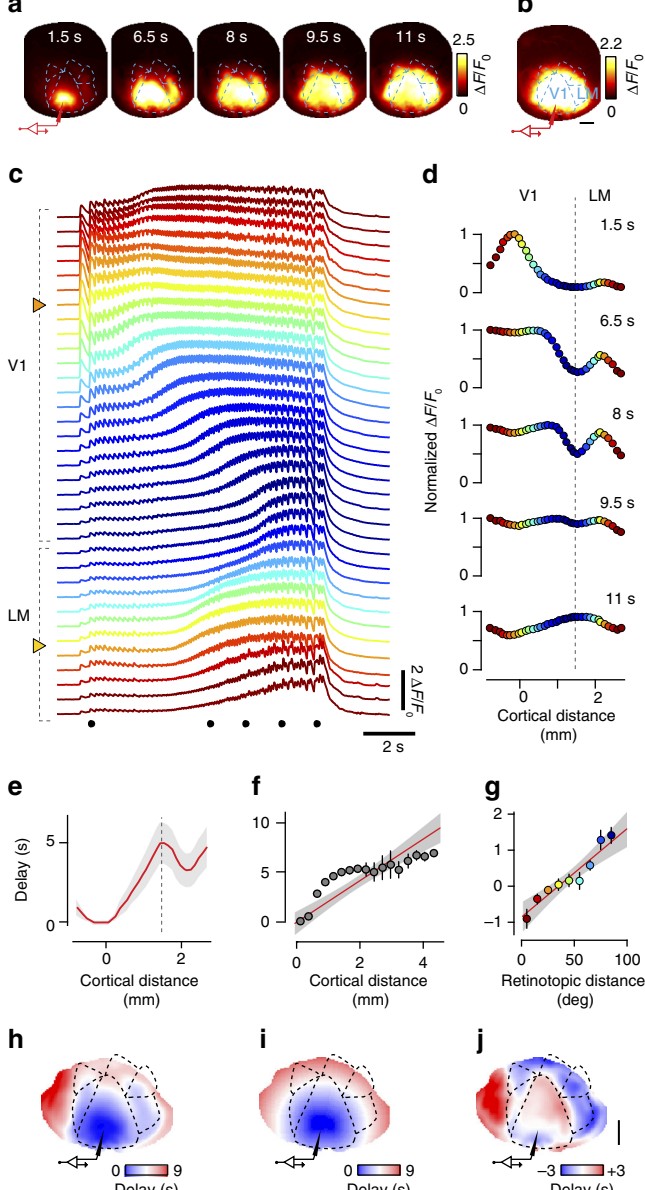

**Fig. 5** Seizure propagation recruits homotopic regions of cortex. **a** Frames imaged in a representative seizure. *Labels* indicate time from seizure onset, and cartoon electrode indicates the site of Ptx injection and LFP recording. **b** Maximum extent of seizure invasion averaged across 14 seizures in this mouse. **c** Single-trial time course along the ROIs in Fig. 4b, for the seizure in **a**. Dots mark the representative times in **a** and **d**. *Arrowheads* mark the focus in V1 and homotopic region in LM. **d** Spatial profile of the seizure in **c**, at representative times, normalized to their maximum. **e** Seizure invasion delay averaged across seizures. *Shaded area* represents s.e.m. **f** Same, averaged across four animals. *Error bars* indicate two s.e.m. *Line:* linear fit, with shaded 95% confidence interval. **g** The residuals of the linear fit in **f** reveal a clear dependence on retinotopic distance from the focus. *Line:* linear fit, with shaded 95% confidence interval. **h** Map of delay to seizure invasion, averaged across 14 seizures in one mouse. **i** Prediction of that map based solely on cortical distance from the focus. **j** Residuals of that prediction. Scale bars are 1 mm

concentric rings (Fig. 5i). The map of residuals instead reveals that it overestimates delays in territories retinotopically close to the focus, especially in higher visual areas (Fig. 5j, *blue negative residual*), and underestimates delays in distal territories in V1 and

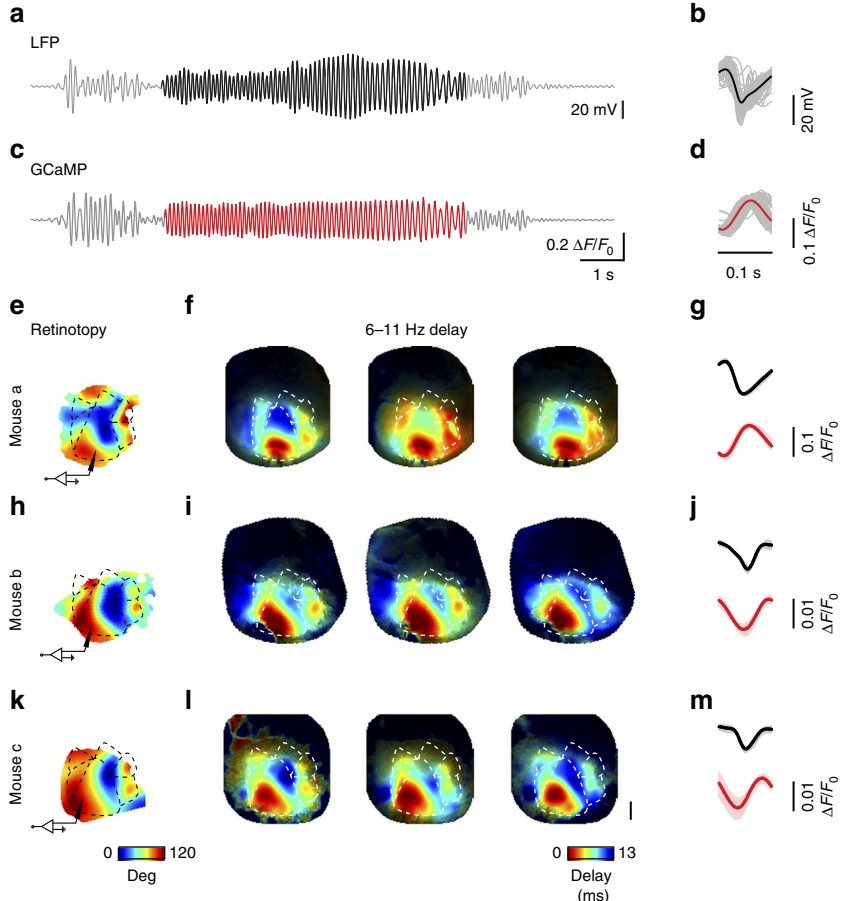

**Fig. 6** Homotopic propagation of 6–11 Hz oscillations during seizures. **a** LFP recording of the seizure in Fig. 5a, bandpass filtered between 6 and 11 Hz. Highlighted in *black* is an epoch of coherent oscillations driven by the focus, used for the analysis in the next panels. **b** Cycle average of the 6–11 Hz oscillation from the unfiltered LFP trace. *Gray traces* show the individual cycles of the oscillation. **c**, **d** Same as **a**, **b**, for the GCaMP fluorescence measured at the focus. Oscillation highlighted in red. **e** Retinotopic maps for an example animal. The *cartoon electrode* indicates the site of Ptx injection and LFP recording, the *dashed line* represents visual areas contours. **f** Oscillatory delay for three example seizures from this example animal. Delay was obtained from the 6–11 Hz Hilbert phase referenced at the focus, averaged across all cycles of the oscillation for a given seizure. **g** Average LFP cycle (*top*) and average GCaMP fluorescence (*bottom*), averaged across all the seizures for this animal. *Shaded area* represents the standard deviation. **h–m** Same as **e–g** for six other seizures in two other mice. In all seizures and all mice, the delay of the oscillation faithfully recapitulates the retinotopic maps. Scale bar: 1 mm

in regions outside the visual areas (Fig. 5j, *red positive residual*). These results indicate that delay to seizure invasion depends approximately linearly on two factors: contiguity, which decreases with distance in cortex, and homotopy, which decreases with distance in visual preference.

**Seizure oscillations propagate along homotopic connectivity**. Homotopic connectivity also determined the timing of individual oscillatory waves during seizures (Fig. 6). Once a territory was invaded by seizure activity, it displayed striking oscillations in the range of 6–11 Hz (Fig. 5c), evident both in the LFP and in the GCaMP fluorescence (Fig. 6a–d). These oscillations started in the focus and spread with the seizure to cover the entire visual cortex and beyond. To isolate these oscillations from the slower spreading activity, we filtered the traces between 6 and 11 Hz, and used the Hilbert Transform to compute maps of amplitude and phase of the oscillation at each time point (Supplementary Movie 3). For a large portion of a seizure duration (for the first $6.6 \pm 0.4$ s), the seizure focus tended to act as a pacemaker for the oscillations: each oscillatory wave rose at the focus and propagated on the surface of the cortex, with homotopic regions oscillating almost in synchrony with activity in the

focus. The spatiotemporal pattern of these oscillations was consistent across oscillations and across seizures (Fig. 6f and g, Supplementary Movie 4) and was closely related to the map of retinotopy (Fig. 6e). Similar results were seen in other mice (Fig. 6h–m). Just like the slow spread of seizure activity, therefore, the faster oscillations within the seizure propagate along the same circuits that support neural signals during normal cortical activity.

Occasionally, however, oscillatory waves showed more complex patterns of propagation such as spiral waves[31] (Supplementary Fig. 8). To characterize the individual oscillations in a seizure (Supplementary Fig. 8a), we took the Hilbert Transform and assigned an amplitude and a phase to each pixel. We then measured the variance of the phase across space (Supplementary Fig. 8b). In most oscillations, the phase had low variance. These were the homotopic waves, where activity oscillated in synchrony at the focus and in homotopic regions, and shortly afterwards (within ~13 ms, Fig. 6) in the remaining regions. In occasional oscillations, instead, the phase had higher variance and described a spiral pattern of activity[31] (Supplementary Fig. 8c): it varied in an orderly manner around a pinwheel point where the amplitude tended to be lower (Supplementary Fig. 8d). Such spiral waves were rare at seizure onset, but became more common towards the

end of the seizure (Supplementary Fig. 8e,f). Once a seizure spread beyond the visual cortex, moreover, the patterns of activity became more complex, and the leading regions in the oscillations were no longer confined to the original focus of the seizure (Supplementary Movie 3).

## Discussion

By exploiting the potential of widefield imaging and of genetically encoded calcium indicators, we were able to study the spatio-temporal evolution of epileptiform activity in the awake cortex and relate this evolution to the underlying pattern of connectivity. Focal picrotoxin injections induced prolonged seizures with clear behavioral correlates, separated by numerous brief interictal events. In the first few hundred milliseconds from their onset, the two types of event were extremely similar: they were standing waves of activity with a fixed spatial profile. Afterwards, however, interictal events quickly subsided while seizures went on to invade much of visual cortex and beyond. Individual 6–11 Hz waves of activity within seizures spread in a similar manner.

Strikingly, all these epileptiform phenomena respected the connectivity of the visual cortex that underlies normal sensory processing. Specifically, the phenomena were shaped by a combination of two factors, one local and dependent on proximity to the focus, and one distal and dependent on homotopy with the focus. Both factors determined the shape of the common spatial footprint characteristic of interictal events and of seizure onset (Fig. 4). Both factors determined the slow spread of the seizures (Fig. 5). Both factors also determined the fast spread of the prominent oscillations that accompany the seizures (Fig. 6).

These local and distal factors likely reflect different mechanisms. Distal propagation must rely on synaptic barrages transmitted from the seizure focus through long-range, excitatory pathways connecting homotopic regions. Local propagation, in addition, can involve collapse of local inhibition[12, 13], although evidence exists for a host of non-synaptic mechanisms such as gap junctions among neurons[11] or astrocytes[10], alterations in the electrochemical gradients of chloride[8, 9] or potassium[7], endogenous electric fields[5, 6] and defective astrocyte calcium signaling[4].

The combination of widefield imaging and genetically encoded calcium indicators represents a substantial advance over electrophysiological recordings performed at a few sites[35–37], and complements the measures that can be obtained with electrode arrays[12, 31]. Widefield imaging has long been recognized as a promising alternative, and has been applied to measure hemodynamic signals[3, 38], voltage-sensitive fluorescent dyes[39], or organic calcium dyes[40]. These techniques, however, provide much weaker signals than genetically encoded calcium indicators. Such indicators can be targeted to specific types of neurons, have high signal to noise ratios and provide a faithful representation of the underlying neural activity, with appropriate spatial and temporal resolution[25, 27, 41]. Indeed, the signal to noise ratio of the calcium indicators used here allowed single-trial analysis without the need to average responses. Recording in awake animals, moreover, avoided the confounding effects of anesthesia.

These differences in methods may explain a difference between our results and those of earlier studies. Whereas we found a strong relationship between the propagation of epileptiform activity and the underlying functional architecture, previous widefield imaging studies found only a weak relationship[39] or no relationship[3, 38]. In addition to the choice of indicators, these discrepancies could be due to differences in the types of functional architecture taken into consideration. Our study, in mouse, asked how propagation depends on the organization of visual cortex into retinotopic areas, whereas the previous studies[3, 38, 39], in carnivores, asked whether propagation depends on the

organization of V1 into regions with similar orientation preference. It is likely that connectivity is stronger between two homotopic regions in different visual areas than between two V1 regions having the same orientation preference.

We did not observe decreases in fluorescence in regions surrounding epileptiform activity, as one might have expected from an "inhibitory surround"[19]. This finding may seem to contradict an earlier study of hemodynamic signals, which were found to decrease in the region around the focus[3]. Neither our study nor that study, however, employed the ideal methods to probe inhibition. Indeed, decreases in hemodynamic signals cannot be interpreted as increases in inhibition, because interneuron activation has the opposite effect[42]. Our study, in turn, measured calcium signals only in excitatory neurons, and these signals reflect mostly supra-threshold activity and may be insensitive to hyperpolarization below threshold. The question of what interneurons do in the region surrounding the focus thus remains open, and can be addressed in future studies by genetically targeting those neurons for widefield imaging[43].

The striking similarity that we found between interictal events and the onset of seizures suggests that the two are mechanistically related. Interictal events have long been recognized as an electrophysiological marker of epilepsy[16], but have no or only subtle clinical correlates[44]. They have been proposed either to prevent seizure initiation[18] or to act as a prelude to seizures[45]. It has, furthermore, been proposed that interictal events dominated by GABAergic signaling give way to glutamatergic pre-ictal events[20]. Our results do not reveal any detectable differences between interictal events and seizures during the first few hundred milliseconds: both are well modeled as standing waves involving the same temporal profile over a fixed spatial territory. It remains to be determined whether a qualitative or quantitative difference in the underlying cellular mechanisms lead to these events dying down (and thus become interictal) or turning into propagating seizures. A plausible distinction is that seizures result from escape from an inhibitory restraint[46]. Indeed, the smaller peak of activity in LM during interictal events is consistent with a barrage of feed-forward activity that fails to recruit a self-sustaining discharge. To test this hypothesis, once again, it would be useful to express the calcium indicator in inhibitory neurons and compare their activity in interictal events and seizures.

The seizures induced in our preparation also bear similarities with seizures observed in patients. Their speed of propagation across the cortex, ~0.5 mm s$^{-1}$, is consistent with measurements obtained with high-density electrode arrays from seizures recorded in humans with cortical epilepsy[12]. Moreover, propagation to LM is consistent with recruitment of more distant areas at short latency. Likewise, oscillatory waves during seizures originated from the leading seizing territory and propagated as traveling waves, as reported in humans[17].

Further, our results demonstrate that cortico-cortical connections that support higher visual processing have an important role in determining the propagation of seizures originating in V1. The involvement of higher-order visual cortex is consistent with seizures seen in occipital epilepsy, which commonly includes both elemental symptoms in specific areas of the visual field and the more complex visual hallucinations[47] that might be expected from the engagement of higher visual areas. It remains an open question whether this homotopic spread to higher visual areas can rest entirely on cortico-cortical connections or if it requires subcortical signaling through common thalamic relays.

Despite these similarities with human focal epilepsy, one should be careful when interpreting the effects of acute application of chemoconvulsants. For instance, small variations in the extent and direction of chemoconvulsant spread could determine differences in the pathways that are recruited in each experiment.

Moreover, the acute disinhibition or overexcitation at the focus seen with chemoconvulsants may be fundamentally different from the mechanisms causing focal epilepsy in humans. Here, we aimed to reduce these risks by minimizing the amount of picrotoxin released at the focus, and indeed, we observed similar effects across experiments. Moreover, we observed similar epileptiform events following the administration of picrotoxin and pilocarpine, compounds with markedly different pharmacological profile. Finally, much of our analysis focuses on seizure propagation into healthy cortical territories, well beyond the pharmacologically-altered focus. These considerations increase the likelihood that the dynamics of the distal propagation we describe could be common to human focal epilepsies.

Our characterization of the spatiotemporal flow of activity in focal cortical seizures may provide a platform for further understanding the underlying mechanisms and for testing therapeutic approaches. In particular, the techniques and results introduced here may help future work on the mechanisms of secondary generalization of partial-onset epilepsy. Secondary generalization sharply reduces the efficacy of surgical removal of the primary lesion[48] and is thus a hallmark of refractory epilepsy, with substantial impact on quality of life and increased risk of mortality[49, 50]. Although secondary generalization ultimately involves subcortical structures[2] it is commonly preceded by invasion of cortical regions[51]. The techniques introduced here may make this process amenable to study, and may provide a platform to identify and test novel therapeutic targets.

## Methods

**Procedures**. All experimental procedures were conducted according to the UK Animals Scientific Procedures Act (1986). Experiments were performed at University College London, under personal and project licenses released by the Home Office following appropriate ethics review.

All data analysis was performed in MATLAB (The MathWorks Inc.).

**Transgenic lines**. *Emx1::Cre; CAG::flex-GCaMP3* mice were generated by crossing the following two transgenic lines: *Emx1-IRES-Cre*, expressing Cre recombinase under the *Emx1* promoter (catalog #005628, The Jackson Laboratory); and a reporter *Ai38-GCaMP3* line (catalog #014538, The Jackson Laboratory), carrying a floxed copy of the *GCaMP3* gene under the strong *CAG* promoter in the *Rosa26* locus. Offspring expressed GCaMP3 in excitatory neurons of the neocortex and hippocampus[24, 28].

*Rasgrf-dCre-CaMKIIa-TTA-TITL-GCaMP6f* triple transgenics[27] were generated by breeding *Rasgrf-dCre, CamKII-tta* and *Ai93(TITL-GCaMP6f)* mice. Three weeks before imaging, TMP was orally administered for 3 consecutive days to trigger dCre-mediated recombination. With this system, *flex-GCaMP6f* recombination was induced only in *Rasgrf* positive neurons, while transcription was further restricted to *CaMKIIa* expressing neurons that also expressed the transcription factor TTA. As a result, GCaMP6 expression was selective to pyramidal neurons located in layers 2/3 and sparsely in layer 5.

Although GCaMP6f allows for the detection of population activity with higher signal to noise ratio than GCaMP3, the cortical activations studied here were so massive that this made little difference. We thus present data obtained with both indicators.

**Surgical procedures**. Using aseptic techniques, animals were chronically implanted with a thinned skull cranial window under isoflurane anesthesia. 0.05 mL rimadyl (Carprofen 5% w/v) was administered before surgery as an anti-inflammatory and analgesic. During the procedure, eyes were protected with ophthalmic gel (Viscotears Liquid Gel, Alcan) and the body temperature maintained around 37 °C. The animal was anesthetized and mounted in a stereotaxic frame. The scalp over the dorsal skull surface was removed to expose the cranium. The left hemisphere parietal bone was thinned using a scalpel blade. Thinning proceeded until the cancellous layer of the skull, which contains blood vessels, was completely removed. A custom-made stainless-steel head plate with a round imaging chamber (8 mm diameter) was then cemented over the thinned area using dental cement (Superbond). Finally, a small drop of UV cement (Norland Products Inc.) was applied inside the imaging chamber and an 8 mm glass coverslip glued onto the thinned skull region. The cement was cured with an UV LED (390 nm, Thorlabs) for 20 s. The animal was allowed to recover at 37 °C in an incubator and provided with rimadyl in the drinking water (0.1 mL in 100 mL) for the 3 days after the surgery.

On the day of LFP recordings, the animal was anaesthetized again and a small skull screw implanted in the contralateral frontal bone, rostral to the cranial window. A silver wire previously soldered to the screw head served as a reference for electrophysiological recordings. After opening a small hole in the cranial window coverslip, a small craniotomy was performed over the target region of V1. The exposed region was finally covered with Kwik-Cast silicone elastomer sealant. The animal was allowed to fully recover before starting the recording session. The typical recording session of epileptiform activity lasted 30–40 min.

**Visual stimulation and retinotopic mapping**. Visual stimuli were presented on three LCD monitors[25], positioned 30 cm from the animal, arranged to span 140° in azimuth and 60° in elevation of the visual field contralateral to the imaged hemisphere. For retinotopic mapping, stimuli were contrast-reversing gratings presented inside a rectangular window. Stimulus duration was 5 s, flickering frequency was 2 Hz, and spatial frequency was 0.03 cycles/degree. To measure the preferred azimuth, the rectangular window was 60° high and 30° wide. To measure the preferred elevation, it was 20° high and 140° wide.

**Electrophysiology**. LFP was recorded with Ag–Cl electrodes in ACSF-filled borosilicate micropipettes (typical tip aperture 2–3 μm, 1–3 MΩ), positioned at 500–600 μm below the dura. To induce epileptiform activity, 10 mM Picrotoxin (Sigma) dissolved in DMSO (Sigma) was added to the recording solution. During these experiments, the pipette was pushed through the dura without applying pressure, to avoid injection of the GABAergic antagonist. In some experiments, we induced epileptiform activity with injections of 100–200 nL of 5M pilocarpine dissolved in DMSO. Because of the viscosity of the solution, the injection pipette was then withdrawn and exchanged with an ACSF-filled borosilicate micropipette to record the LFP. The pressure inside injection pipettes was measured with a differential manometer and kept below 200 mbar. The resistance of the micropipette was monitored to ensure that insertion in the brain did not clog the tip. LFP signals were amplified 1000-fold and high-pass filtered above 0.1 Hz via a Multiclamp 700B differential amplifier (Molecular Devices). Data were digitized at 10 kHz with a Blackrock acquisition system and software. The exposure signal from the camera, the TTL signals from the visual stimulation software and the screen photodiode signal were recorded at the same time to synchronize electrophysiological, imaging and visual stimulation data.

**LFP analysis**. To identify the start times of interictal events and seizures from the LFP recording we devised a simple supervised event detection method. The LFP trace was resampled at 100 Hz; the start of each event was assigned to the local maxima of the first derivative of the signal that exceeded three standard deviations from baseline; the end of each event was identified when its low frequency envelope returned to baseline. Finally, the results of the automatic detection were inspected visually and events classified as interictal events or seizures. Missed events were manually selected and artifacts removed.

Wavelet scalograms of the LFP signal were computed using the wavelet toolbox provided by Torrence and Compo (atoc.colorado.edu/research/wavelets). Power spectra and coherence of LFP and fluorescence signals were calculated using the Chronux Toolbox[52] (http://chronux.org/).

**Widefield calcium imaging**. Imaging experiments were performed under a custom built tandem-lens epifluorescence macroscope[25, 53]. Excitation light at 480 nm was provided by a blue LED light (465 nm LEX2-B, Brain Vision, band-passed with a Semrock FF01-482/35 filter) and diverted via a dichroic mirror (FF506-Di03, Semrock) into a Leica 1.0× Plan APO (M series, part #10450028) or 1.6× Plan APO objective (M series, part #10450029), which focused the light on the sample. The collected fluorescence was reflected by a second dichroic mirror (FF593-Di03, Semrock), passed through an emission filter (FF01-543/50-25, Semrock), and focused by a second Leica 1.0× Plan APO objective onto the sensor of a sCMOS camera (pco.edge, PCO AG). The camera was controlled by a TTL external trigger synchronized with the visual stimulation. The image acquisition rate was 50 Hz for retinotopic mapping and 70 Hz during imaging of epileptiform activity (except for experiments in the pilocarpine model, imaged at 50 Hz). The nominal spatial resolution of the system was 200 pixel/mm. Imaging was conducted in 20 s long sweeps, interleaved with ~10 s pauses needed to save the data to disk.

**Analysis of imaging movies**. Widefield movies where first registered using a discrete Fourier transform based subpixel image registration algorithm[54]. The retinotopic mapping and epileptiform activity images where then aligned using the brain vasculature as a reference. Affine transformation was used to find the best alignment. Then we calculated $\Delta F/F_0$ movies, where the signal for each pixel $x$ at time $t$ was obtained as

$$\frac{\Delta F(t,x)}{F_0} = \frac{F(t,x) - F_0(x)}{F_0(x)}$$

Where $F_0$ was computed as the twentieth percentile of each imaging sweep. Finally, a mask was manually drawn to segment out the visible cortex form the edges of the implant.

The map of retinotopy was obtained as previously described[25]. In short, the reversing gratings elicited periodic neural responses that oscillate at twice the frequency of reversal[55]. For each pixel, we measured the power of these second harmonic responses using the Fourier transform of the fluorescent trace. We then fitted a Gaussian position-tuning curve to the amplitude of the responses to visual stimulation at four horizontal positions. The standard deviation for the Gaussian curve model was the one that minimized the least squared error of the fit across pixels.

To localize the cortical initiation site of each event, calcium signals were aligned by the start of the electrographic discharge. For each event, pixels were included if they showed $\Delta F/F$ increases greater than 60% of the peak $\Delta F/F$ reached in the same class of events. The center of mass of the resulting cortical area, calculated over the first 500 ms, was then taken as the site of initiation of epileptiform activity. The average retinotopic position of the focus across animals was $93 \pm 16$ degrees, confirming our ability to target the medial region of V1. We restricted the analysis to events that were contained for the most part in an imaging sweep. In particular, we included seizures that were imaged for at least 6 s after their start.

To assess the slow propagation of seizures, we defined criteria to evaluate when a cortical territory was invaded by the seizing activity. The distribution of maximum florescence levels during seizure was bimodal (Supplementary Fig. 7). We considered recruited to the seizure the pixels that exceeded 30% of the peak fluorescence recorded at the focus, which was a good threshold to separate the population of pixels with the highest fluorescence change (Supplementary Fig. 7). We then calculated the time it took for a seizure to invade each recruited pixel, measured from the electrographic seizure onset. We called this time 'delay to seizure invasion', and we measured it as the time point at which the fluorescence at any recruited pixel reached 60% of its maximum.

**Standing wave model**. We modeled the fluorescence movies of epileptiform events as standing waves (Fig. 3e), i.e. as the product of a global time course and a single map (an image)

$$\text{Movie} = \text{Timecourse} \times \text{Map} + \text{Residuals}$$

To fit this model (i.e. to minimize the residuals), we used singular value decomposition (SVD), which returns the best least square approximation of a matrix as the sum of standing waves. We obtained the time course and map of the separable model from the first row and first column returned by the SVD.

**Modeling of the profile of interictal events**. We modeled the retinotopic profile of activation elicited by interictal events as the dot product of two functions, both dependent on the cortical position $x$ of each ROI.

The first function depends on homotopic connectivity, and is a Gaussian falling off with the distance between the retinotopic position $R(x)$ of point $x$ and the retinotopic position of the focus, $R_{\text{focus}}$:

$$H(x) = e^{-\frac{(R(x)-R_{\text{focus}})^2}{2\sigma^2}}$$

The second function depends on contiguous spread, and is a Gaussian-like function with the numerator raised to the fourth power to give it a flat top as it falls off with the cortical distance from the epileptic focus:

$$C(x) = (a-b)e^{-\frac{(x-x_{\text{focus}})^4}{\lambda^4/\ln(2)}} + b$$

When so parametrized, $\lambda$ corresponds to the half-width half-maximum of the function. We fitted the parameters $\sigma$, $\lambda$, $R_{\text{focus}}$, $x_{\text{focus}}$, $a$, $b$ by least square minimization.

**Phase analysis of cortical oscillations**. Movies of cortical seizures were first bandpass filtered between 6 and 11 Hz. Then we computed the analytical representation of the fluorescence time course for each pixel, using the Hilbert transform (MATLAB function 'hilbert', Signal Processing Toolbox). The analytical signal $S_a(t)$ is a representation of a real valued signal $f(t)$ in the complex space:

$$S_a(t) = f(t) + iH(f)(t) = A(t)e^{i\varphi(t)}$$

Its real part is $f(t)$, the fluorescence signal, while its imaginary part is $H(f)(t)$, the Hilbert transform of $f(t)$. When expressed in polar form, the analytical signal $S_a(t)$ has a straightforward interpretation: the modulus $A(t)$ represents the instantaneous amplitude, or envelope, of the oscillation; the angle $\varphi(t)$ represents the instantaneous phase of the oscillation.

**Behavioral data**. The eye contralateral to the imaged cortex was imaged with a monochrome CCD camera (DMK 21BU04.H, The Imaging Source), equipped with a macro zoom lens (MVL7000—18–108 mm EFL, f/2.5, Thorlabs). Illumination to

the eye was provided by two infrared LEDs (850 nm, Mightex, powered by an SLC-SA/AA LED controller, Mightex). An infrared filter (700 nm high-pass, #092, The Imaging Source) was applied to the camera objective to shield it from stray light. Eye imaging movies were processed with custom software to extract eye movements, pupil dilations and blinks.

The running behavior was measured using a rotary encoder (Kübler 05.2400.1122.0100) that tracked the revolutions of the cylindrical treadmill. The angular displacement was first converted into angular speed and then into linear speed based on the diameter of the treadmill.

Both the rotary encoder signal and the frame time stamps from the eye-tracking camera were acquired using a data acquisition board (National Instruments, PCIe-6323). The same board acquired a copy of the exposure signal from the imaging camera, the TTL signals from the visual stimulation software and the screen photodiode signal: such signals were used to align imaging, electrophysiological and behavioral recordings.

**Statistical information**. Data were tested for normality before choosing the appropriate statistical test. All the statistical tests used were 2-tailed test. The significance of linear regression models was tested with the F-statistic against the null hypothesis of a constant model. $p = 0.05$ was assumed as the threshold for statistical significance.

**Code availability**. All the relevant code is available from the authors upon request. Representative code can be found at www.ucl.ac.uk/cortexlab/data.

**Data and code availability**. All the relevant data are available from the authors upon request. An example data set can be found at www.ucl.ac.uk/cortexlab/data.

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

## Acknowledgements

We thank Michael Okun and Daisuke Shimaoka for assistance in experimental set-up and data analysis, and Charu Reddy for animal husbandry. This work was supported by the Wellcome Trust (095669 and 095580), the Medical Research Council, and Epilepsy Research UK (grant F1401 to R.C.W.). L.F.R. was funded by the Wellcome Trust PhD program in Neuroscience at UCL. M.C. holds the Glaxo-SmithKline/Fight for Sight Chair in Visual Neuroscience.

## Author contributions

All authors designed the experiments. L.F.R. and R.C.W. performed experiments. L.F.R. analyzed data, made the figures, and drafted the paper. All authors edited the paper.

## Additional information

**Competing interests:** The authors declare no competing financial interests.

