## [Peer Review File · Nature Communications]

Reviewers' Comments:

Reviewer #1 (Remarks to the Author):

In this paper, the authors describe the propagation of epileptiform events in mouse visual cortex using a combination of LFP recording and imaging of genetically encoded calcium indicators. Analysis of the spatiotemporal evolution of the calcium signal during ictal and interictal events showed similar patterns in the initial response, but ictal activity moved to higher-order visual areas more quickly than would be expected by lateral spread alone. The authors then showed that this distal activity in area LM originated in a retinotopically matched region to the ictal focus in V1. This was taken as evidence that the propagation of seizures in cortex follows the patterns of functional connectivity that underlie normal visual processing.

A major strength of this paper is the utilization of multiple techniques (electrophysiology and widefield, genetically-encoded calcium imaging) in an awake rodent model to address questions of propagation of seizure-like events in an elegant manner. The variants of GCaMP used in this study provide a high signal to noise ratio and enough temporal resolution to capture some oscillatory characteristics of the cortical activity on a trial-to-trial basis. This allows for analyses of timing and phase that would otherwise require large electrode arrays and their associated drawbacks.

However, I have major concerns regarding data interpretation and relevance. Although the authors developed a nice method to study the signal propagation, the manuscript is very descriptive and does not address the circuit mechanisms of signal propagation. The strength of the paper is the method and I encourage the authors to emphasize this part rather than focusing on how they shed light on mechanisms of epileptic seizure generalization – which is an overstatement given that the model they use is not an epilepsy model and it is unclear how specific the results are to the pharmacological acute model of hyperactivity that they chose to use here.

I have some minor questions regarding the role of the mouse's behavioral state in the observed activity, and about how to determine the mechanisms underlying homotopic spread of seizures.

Major concerns:

In the Discussion section the authors claim that their “characterization of the spatiotemporal flow of activity in epileptic discharges provides a platform for further understanding the mechanisms of epilepsy and for testing therapeutic approaches”.

The acute pharmacological seizure model used in this study is not a model of epilepsy, and therefore the authors can not claim that they characterized the “epileptic discharges”. This wording would be appropriate for an animal model of epilepsy and not an animal model of acute seizures. Indeed, the cellular and circuit mechanisms of epileptic discharges and of ictal propagation can be very different from those of acute pharmacologically induced seizures. Therefore, this study does not allow “understanding of the mechanisms of epilepsy”. The authors should remove this statement. The authors can emphasize that this study introduces an elegant method to study propagation that could be relevant in for future studies of the spread of seizures in animal models of epilepsy.

“In particular, the techniques and results introduced here may help shed light on the mechanisms of secondary generalization of partial-onset seizures”.

This is an overstatement. The techniques used in this study may be further used to shed light on the propagation of epileptic seizures in animal models of epilepsy. However, the results provided here do not shed light on how epileptic seizures would spread in an animal model of epilepsy. It is unclear to what extent are the picrotoxin-induced events described in this paper reflect epileptic seizures rather than a pharmacologically induced network hyperactivity in a normal brain.

The large amplitude, rhythmicity, and rate of spread shown here seem consistent with epileptic seizures, but in many ways the behavioral response is relatively mild. The authors describe a consistent increase in pupil diameter and running speed, but no other behavioral measures. It is not evident if any events recorded generalize beyond visual cortex.

My other concern with the induction method is that it is unclear how much variability is present with the amount of picrotoxin release locally and its area of effect. Pipettes were targeted to layer 5 of V1, in order to take advantage of connectivity of L5 cells across cortical laminae. While the amount of picrotoxin was designed to be minimal, the extent of its effect is unclear, and could recruit different pathways depending on amount and direction of spread. While the dendrites of L5 pyramidal cells span all layers of cortex, the axons primarily target subcortical regions (basal ganglia, thalamus...). Does the local spread of activity in response to picrotoxin depend on diffusion, dendritic spiking, or subcortical relays? Would similar responses be observed if superficial layers 2/3 were targeted, as they project primarily intracortically?

Minor concerns:

-Given that the behavioral correlates of ictal events in this preparation are pupil dilation, eye movements, and increased running, to what extent are the data from these periods confounded by normal changes in visual processing accompanying fast changes in gaze, or generally increased arousal?

-What mechanisms account for the delay in homotopic spread during ictal events? As noted, this must require long-range synaptic activity, but peak signal in LM is seen a number of seconds after an event starts in V1 (so much longer than a few synaptic delays). I am curious about whether the authors think this connection is primarily cortical or whether it requires subcortical signaling, and what is needed to recruit these pathways, although that may be beyond the scope of the current study.

Reviewer #2 (Remarks to the Author):

The paper uses state of the art techniques to monitor the activity of cerebral cortex during seizures, and their spread from the focus. The principal originality of the study is the high-resolution spatial imaging of the seizure using wide field calcium imaging, which is very useful information. I have a few major remarks.

1. It is not clear why the authors speak of "functional connectivity". It seems that the mouse V1 is precisely not a good place to look for functional connectivity since it has no orientation maps and columns, very far from all the functional architecture that one can see in higher mammals. They show a lateral spread, presumably mediated by axon collaterals, fine. They also show long range interactions and subsequent spread, and this is presumably mediated by long-range axonal connectivity. So why speak of functional connectivity? It seems to me that the paper reports the effect of short-range and long-range connections, and not of functional connectivity. The authors should change this interpretation, or provide more evidence.

2. The authors should give more credit to previous work. The investigation of epilepsy in V1 is a very unusual topic, especially following induction of an artificial focus using picrotoxin. This experimental model is the same as proposed by Viventi et al. (Nat Neurosci 2011), they studied epileptic discharges also in V1, also triggered by picrotoxin, and also using a system for high-resolution spatial imaging. The main difference was that this prior study was on cat. The authors should give much more credit to these studies. In fact, in vivo recordings of cortical seizures following a cortical focus induced using GABA(A) antagonists, was investigated by various groups, such as Gloor and Steriade. In find it very disturbing that this work is not even cited, as if this experimental model was invented here - a non-aware reader will get this wrong impression, and this is clearly not correct.

3. Regarding Viventi et al., their finding of a spiral wave during the seizure, in cat V1, where the functional architecture is very well established, seems to contradict the idea that the seizure follows functional links. Why would it be the case in mice while it is obviously not the case in cats? This further suggest that the "functional" interpretation is not correct.

If this is true, then it is problematic for the paper, because its novelty is not clear. Besides being technically impressive, the experiments show something we already know from lower resolution studies, that seizures have a local spread, and a global spread. The authors should really stress what is the new information that we learn from the paper.

4. Regarding the absence of inhibitory surround, and the criticism that previous studies did not look at inhibition, the authors may want to comment on the recent paper of Dehghani et al. (Sci Reports 2016) about the breakdown of excitatory-inhibitory balanced activity in human seizures, with apparent potentiation of inhibitory neuron activity during the seizure. Supposing that such a potentiation of inhibitory neuron activity is also present here (and not seen by the calcium imaging), would that change your interpretations? Do you have any evidence for or against it?

Minor remarks:

1. The authors seem to call a 6-11 Hz oscillation as "fast" but this is within the sleep spindle or alpha frequency range, which are usually considered as "slow" oscillations (while "fast oscillation" usually refers to gamma or ripple oscillations). This should be reworded.

Finally, a question, are the Gcamp6 imaging results in full agreement with the LFP recordings? Since LFPs are mostly generated by synaptic activity, and the calcium imaging signal by spiking activity, wouldn't we expect differences? Is there a difference in this respect between "normal" activity and seizure? Can you comment?

Reviewer #3 (Remarks to the Author):

In this manuscript, Rossi et al. present their findings that pharmacologically induced epileptic seizures propagate not only locally but also through longer-range projections to homotopic regions within visual cortex. I have some suggestions below, but I generally found the data novel and convincing, and am in support of publication.

Primary concerns/suggestions:

1) The analyses focused exclusively on distal propagation from V1 to LM. However, work from the Burkhalter lab and others have shown that region PM also receives extensive retinotopic input from V1 and would be expected to show distal propagation of seizures from the V1 locus.

It would be great to see the authors extend their analysis in Figures 4 and 5 to region PM. This could be done with the existing data, simply extending the line of fixed vertical preference from $PM > V1 > LM$. If they do not see the same effect in PM, there should be some treatment of the PM/LM differences in the discussion.

2) I was somewhat confused by Figure 6. Is the point that locations that share retinotopy with the seizure initiation point have low latency delay? (If this is the case, showing seizure initiation points would help make this point clear - see minor point #4). Or alternatively, is the point that low latency delay is always mapped to horizontal retinotopy regardless of the initiation site? (This doesn't make much sense to me, but it is unclear from the figure). In any case, this analysis should be presented more clearly in the text/figure to avoid confusion.

Minor concerns/suggestions:

- 1) Grammatical error in line 15 of abstract.
- 2) References 22/30: duplicate citation
- 3) Typo in Fig 1i: red triangles > blue triangles
- 4) It would be helpful to indicate the location of the electrode tip (seizure initiation point), particularly in Fig 3 and 6.
- 5) The methods section (and acknowledgements) refers to histology and confocal imaging, but I couldn't find any histology in the paper.
- 6) More details on the imaging system should be included in the "widefield calcium imaging" section of the methods. It wasn't clear to me whether the authors used a tandem lens or objective-based imaging system. The specs and part numbers on the major optics should also be included.
- 7) Two typos in lines 548 and 549: where > were.

We thank the reviewers for their useful comments and criticisms. To address them we made the following main changes to the paper:

- We changed the title to clarify that the paper concerns cortical focal seizures and summarize the main results
- We clarified what we mean by “functional circuits”, and removed references to “functional connectivity”.
- We added experiments with a different pharmacological model to show that it leads to similar results
- We discussed the limitations of studying focal seizures using pharmacological methods.
- We added a figure showing that our induced seizures generalize beyond visual cortex
- We added a comparison between pupil behaviors during epileptic and control conditions.
- We provide more thorough comparison with the existing literature as suggested by the reviewers
- We added an explanation of the occasional spiral waves that we observed, and a video of these waves

Below are detailed responses to the reviewer comments.

Reviewer 1

In this paper, the authors describe the propagation of epileptiform events in mouse visual cortex using a combination of LFP recording and imaging of genetically encoded calcium indicators. Analysis of the spatiotemporal evolution of the calcium signal during ictal and interictal events showed similar patterns in the initial response, but ictal activity moved to higher-order visual areas more quickly than would be expected by lateral spread alone. The authors then showed that this distal activity in area LM originated in a retinotopically matched region to the ictal focus in V1. This was taken as evidence that the propagation of seizures in cortex follows the patterns of functional connectivity that underlie normal visual processing.

A major strength of this paper is the utilization of multiple techniques (electrophysiology and widefield, genetically-encoded calcium imaging) in an awake rodent model to address questions of propagation of seizure-like events in an elegant manner. The variants of GCaMP used in this study provide a high signal to noise ratio and enough temporal resolution to capture some oscillatory characteristics of the cortical activity on a trial-to-trial basis. This allows for analyses of timing and phase that would otherwise require large electrode arrays and their associated drawbacks.

We thank the reviewer for this effective summary of our methods and findings. To this summary we would like to add that our methods revealed that interictal event and a propagating seizure start by being extremely similar: they are both standing waves. The analysis in terms of standing vs. travelling waves is a key point in our analysis and we have now made this more explicit by stating it in the title.

However, I have major concerns regarding data interpretation and relevance. Although the authors developed a nice method to study the signal propagation, the manuscript is very descriptive and does not address the circuit mechanisms of signal propagation. The strength of the paper is the method and I encourage the authors to emphasize this part rather than focusing on how they shed light on mechanisms of epileptic seizure generalization – which is an overstatement given that the model they use is not an epilepsy model and it is unclear how specific the results are to the pharmacological acute model of hyperactivity that they chose to use here.

We agree with the reviewer. The strength of our work is in the techniques we used and in the quantitative summary of the results by simple mathematical descriptions such as the standing wave model. The only claim that we can make as to mechanism concerns the patterns of connectivity: the propagation that we observe is highly consistent with the expectation from homotopic cortical connectivity. We have extended the Discussion to emphasize the limitations of the conclusions that can be drawn from our study, and we have made sure that there are no overstatements that go beyond these conclusions.

I have some minor questions regarding the role of the mouse’s behavioral state in the observed activity, and about how to determine the mechanisms underlying homotopic spread of seizures.

Major concerns:

In the Discussion section the authors claim that their “characterization of the spatiotemporal flow of activity in epileptic discharges provides a platform for further understanding the mechanisms of epilepsy and for testing therapeutic approaches”.

The acute pharmacological seizure model used in this study is not a model of epilepsy, and therefore the authors can not claim that they characterized the “epileptic discharges”. This wording would be appropriate for an animal model of epilepsy and not an animal model of acute seizures. Indeed, the cellular and circuit mechanisms of epileptic discharges and of ictal propagation can be very different from those of acute pharmacologically induced seizures. Therefore, this study does not allow “understanding of the mechanisms of epilepsy”. The authors should remove this statement. The authors can emphasize that this study introduces an elegant method to study propagation that could be relevant in for future studies of the spread of seizures in animal models of epilepsy.

“In particular, the techniques and results introduced here may help shed light on the mechanisms of secondary generalization of partial-onset seizures”.

This is an overstatement. The techniques used in this study may be further used to shed light on the propagation of epileptic seizures in animal models of epilepsy. However, the results provided here do not shed light on how epileptic seizures would spread in an animal model of epilepsy. It is unclear to what extent are the picrotoxin-induced events described in this paper reflect epileptic seizures rather than a pharmacologically induced network hyperactivity in a normal brain.

These comments concern the wording of the last paragraph of Discussion, and specifically of two sentences in that paragraph. To address them, we modified both sentences.

In the first sentence, we have made the claim more specific to focal epilepsy and more tentative. The new sentence reads “Our characterization of the spatiotemporal flow of activity in focal epilepsy may provide a platform for further understanding the underlying mechanisms and for testing therapeutic approaches.”

As for the second sentence, it must have been ambiguous, because it led to a misunderstanding. The reviewer read it as a claim that our model of epilepsy shows secondary generalization. We simply meant to say that future studies of secondary generalization may find our techniques useful. To avoid such misunderstandings, the sentence now reads: “In particular, the techniques and results introduced here may help future work on the mechanisms of secondary generalization of partial-onset seizures.”

The large amplitude, rhythmicity, and rate of spread shown here seem consistent with epileptic seizures, but in many ways the behavioral response is relatively mild. The authors describe a consistent increase in pupil diameter and running speed, but no other behavioral measures. It is not evident if any events recorded generalize beyond visual cortex.

We thank the reviewer for this comment. As we now make clearer in the paper, we did observe seizures that generalized beyond visual cortex, to the entire imaged portion of the left hemisphere. We now show this in a new Supplementary Figure (Suppl. Fig. 4). Moreover, we added a comparison between the behavioral correlates of seizures and the behavior of the animal in control condition (Suppl. Fig 1), to show that the pupil dilations during seizures are indeed extreme. Finally, we did observe behavioral correlates typical of secondary generalization (tonic limb movements, tail flicking) and we now comment on this in the paper.

My other concern with the induction method is that it is unclear how much variability is present with the amount of picrotoxin release locally and its area of effect. Pipettes were targeted to layer 5 of V1, in order to take advantage of connectivity of L5 cells across cortical laminae. While the amount of picrotoxin was designed to be minimal, the extent of its effect is unclear, and could recruit different pathways depending on amount and direction of spread. While the dendrites of L5 pyramidal cells span all layers of cortex, the axons primarily target subcortical regions (basal ganglia, thalamus...). Does the local spread of activity in response to picrotoxin depend on diffusion, dendritic spiking, or subcortical relays? Would similar responses be observed if superficial layers 2/3 were targeted, as they project primarily intracortically?

We agree that variability is a potential concern, and to address it we have videos showing how the results were similar across mice. As for the factors listed by the reviewer as potentially playing a role, we have now added a

new paragraph in Discussion (the paragraph before the last) where we list them using similar words as those used by the reviewer.

Minor concerns:

-Given that the behavioral correlates of ictal events in this preparation are pupil dilation, eye movements, and increased running, to what extent are the data from these periods confounded by normal changes in visual processing accompanying fast changes in gaze, or generally increased arousal?

Pupil dilations, eye movements, and running are indeed three factors that can increase activity in primary visual cortex. The reviewer wonders whether changes in those factors could itself explain increased activity in cortex. However, the increase they cause is minuscule compared to the 100x increase in activity seen during epileptiform events. This can be seen, for instance, in an excellent review paper that appeared recently [McGinley, M.J., Vinck, M., Reimer, J., Batista-Brito, R., Zaghera, E., Cadwell, C.R., Tolias, A.S., Cardin, J.A., and McCormick, D.A. (2015). Waking State: Rapid Variations Modulate Neural and Behavioral Responses. *Neuron* 87, 1143-1161.].

-What mechanisms account for the delay in homotopic spread during ictal events? As noted, this must require long-range synaptic activity, but peak signal in LM is seen a number of seconds after an event starts in V1 (so much longer than a few synaptic delays). I am curious about whether the authors think this connection is primarily cortical or whether it requires subcortical signaling, and what is needed to recruit these pathways, although that may be beyond the scope of the current study.

The question of what determines the slow propagation of epileptic activity is a very important one, but our measurements do not allow us to answer it except to note that the propagation speed that we observe is comparable to that observed in human epilepsy. We have now added some thoughts in Discussion, where we point out the possible involvement of thalamic relays, and we indicate that this could be a fruitful area of further study.

Reviewer 2

The paper uses state of the art techniques to monitor the activity of cerebral cortex during seizures, and their spread from the focus. The principal originality of the study is the high-resolution spatial imaging of the seizure using wide field calcium imaging, which is very useful information. I have a few major remarks.

1. It is not clear why the authors speak of "functional connectivity". It seems that the mouse V1 is precisely not a good place to look for functional connectivity since it has no orientation maps and columns, very far from all the functional architecture that one can see in higher mammals. They show a lateral spread, presumably mediated by axon collaterals, fine. They also show long range interactions and subsequent spread, and this is presumably mediated by long-range axonal connectivity. So why speak of functional connectivity? It seems to me that the paper reports the effect of short-range and long-range connections, and not of functional connectivity. The authors should change this interpretation, or provide more evidence.

We agree that "functional connectivity" is often used in the field in a vague or inappropriate way, but in our case, we believe it was used properly, to indicate "connectivity that has been confirmed functionally". We characterized the retinotopic organization of V1 and higher visual areas (which receive their visual input from V1) prior to eliciting seizures. The relative retinotopy of V1 and higher visual areas is therefore precisely a reflection of function (processing of visual information) and connectivity (the underlying anatomical and physiological processes). This said, because the term "functional connectivity" can be interpreted in multiple ways, we have removed it from the paper, and instead we refer to it as "connectivity" or "homotopic connectivity" where that is more appropriate.

We have also amended the Discussion to explain that although visual cortex in the mouse does not exhibit some features seen in primates and carnivores, this does not mean that there is no functional connectivity. Indeed, the reviewer's point that the relative retinotopy of V1 and LM merely reflects "short-range and long-range connections" could equally be made about any other form of functional connectivity in the literature.

2. The authors should give more credit to previous work. The investigation of epilepsy in V1 is a very unusual topic, especially following induction of an artificial focus using picrotoxin. This experimental model is the same as proposed by Viventi et al. (Nat Neurosci 2011), they studied epileptic discharges also in V1, also triggered by picrotoxin, and also using a system for high-resolution spatial imaging. The main difference was that this prior study was on cat. The authors should give much more credit to these studies. In fact, in vivo recordings of cortical seizures following a cortical focus induced using GABA(A) antagonists, was investigated by various groups, such as Gloor and Steriade. In find it very disturbing that this work is not even cited, as if this experimental model was invented here - a non-aware reader will get this wrong impression, and this is clearly not correct.

We thank the reviewer for pointing out possible weaknesses in our scholarly treatment of the literature, but we struggle to see how such weaknesses could have been "highly disturbing". In our previous version we did cite the work of Viventi et al in reference to the spiral waves dynamics they reported. We have now added a citation to them in an earlier part of the paper, to emphasize that they also used picrotoxin to trigger seizures. Moreover, we also cite older work by Steriade et al. (which Viventi et al did not feel the need to cite, by the way) to acknowledge their extensive study of neocortical seizures triggered by this GABA_A receptor blocker.

Regarding the novelty relative to the work by Viventi et al., we disagree with the reviewer: they did not use "high resolution imaging". Instead they used surface electrode arrays, which have such low resolution that they could not even reveal the most striking feature of cat visual cortex: its huge orientation domains.

As for epilepsy in V1 being an unusual topic, we would like to stress that occipital lobe epilepsy affects 5-10% of people with epilepsy (that is, up to 0.1% of the entire population, Sveinbjornsdottir & Duncan *Epilepsia* 1993, 34:493-521). It has major consequences for affected individuals because surgical resection is hardly an option as it leads to permanent visual defect.

3. Regarding Viventi et al., their finding of a spiral wave during the seizure, in cat V1, where the functional architecture is very well established, seems to contradict the idea that the seizure follows functional links. Why would it be the case in mice while it is obviously not the case in cats? This further suggest that the "functional" interpretation is not correct.

It is hard to compare our study to that of Viventi et al, as they did not characterize the functional architecture of V1. Their data indicate neither the border of V1 and other areas nor the outlines of orientation columns. Because they did not measure the underlying functional architecture, it is impossible to relate their measurements to the underlying functional architecture.

Moreover, it is hard to compare our methods with those of Viventi et al. as that study did not provide a detailed description of how they induced seizures, nor did they give details on the duration or phenomenology of such seizures. The only sentence in their paper is '*The drug was placed directly on the brain, adjacent to the electrode array on the frontal-medial corner*'. Therefore, it is unclear if they were also triggering focal seizures, or if the activity they studied was instead triggered by diffuse cortical disinhibition throughout the sampled area. In contrast, we deliberately applied the chemoconvulsant only via the LFP electrode and waited until discrete interictal discharges and seizures arose, invading neighboring areas of cortex.

Finally, Viventi et al. recorded from a restricted area of cortex, dictated by the size of the electrode array, whilst we recorded from an entire hemisphere, with multiple visual areas. This allowed us to relate seizure propagation to functional organization from a more global perspective.

Although these points emphasize the differences between Viventi et al. and our study, there were also similarities. We also recorded spiral wave dynamics, especially evident in the early and late phase of the seizures. We have added Suppl. Fig 7 and Suppl. Movie 3 to characterize this behavior. The spiral waves reported by Viventi et al. could be akin to the ones we see in the first phase of the seizure, which remain quite local in V1. In fact, the seizures recorded in Viventi et. al appear quite short, although we could only gauge this from the example provided in figure, because they do not discuss seizure duration at all. Nonetheless, it seems that the phase relationship between matching retinotopic patches of visual areas is partially preserved during these early waves as well.

If this is true, then it is problematic for the paper, because its novelty is not clear. Besides being technically impressive, the experiments show something we already know from lower resolution studies, that seizures have a local spread, and a global spread. The authors should really stress what is the new information that we learn from the paper.

We have now changed the title of the paper to emphasize that the main findings concern cortical focal epilepsy, and the fact that epileptiform events start as standing waves and propagate homotopically. There are also a number of other results relating both the fast propagation of epileptiform activity and the slow spread of this activity to the underlying connectivity. Specifically, the finding that distal spread reaches retinotopically similar areas before invading more proximal areas is a novel and non-trivial result. If anything, the existing literature (mostly work by Schwartz) indicates the opposite. And the paper by Viventi et al., which the reviewer mentions extensively, has no bearing on these findings.

4. Regarding the absence of inhibitory surround, and the criticism that previous studies did not look at inhibition, the authors may want to comment on the recent paper of Dehghani et al. (Sci Reports 2016) about the breakdown of excitatory-inhibitory balanced activity in human seizures, with apparent potentiation of inhibitory neuron activity during the seizure. Supposing that such a potentiation of inhibitory neuron activity is also present here (and not seen by the calcium imaging), would that change your interpretations? Do you have any evidence for or against it?

There must be a misunderstanding, as our paper does not criticize previous studies in any way. Our methods do not allow us to distinguish excitation from inhibition. However as we point out in the paper, our methods could be used to address these questions: "it would be useful to express the calcium indicator in inhibitory neurons".

Minor remarks:

1. The authors seem to call a 6-11 Hz oscillation as "fast" but this is within the sleep spindle or alpha frequency range, which are usually considered as "slow" oscillations (while "fast oscillation" usually refers to gamma or ripple oscillations). This should be reworded.

We thank the reviewer for pointing this out. We have now replaced "fast" with "6-11 Hz".

Finally, a question, are the Gcamp6 imaging results in full agreement with the LFP recordings? Since LFPs are mostly generated by synaptic activity, and the calcium imaging signal by spiking activity, wouldn't we expect differences? Is there a difference in this respect between "normal" activity and seizure? Can you comment?

The relationship between LFP and other measurements of neural activity is an important area of research, but it is secondary to the present study. We did observe differences between GCaMP signals and LFP signals, mostly in that the former are much slower than the latter (as expected by the dynamics of the sensor). Our concern was whether the calcium signal would be fast enough to follow the 6-11 Hz oscillations, and the power spectra and coherence measures confirm that they do.

Reviewer 3

In this manuscript, Rossi et al. present their findings that pharmacologically induced epileptic seizures propagate not only locally but also through longer-range projections to homotopic regions within visual cortex. I have some suggestions below, but I generally found the data novel and convincing, and am in support of publication.

Primary concerns/suggestions:

1) The analyses focused exclusively on distal propagation from V1 to LM. However, work from the Burkhalter lab and others have shown that region PM also receives extensive retinotopic input from V1 and would be expected to show distal propagation of seizures from the V1 locus. It would be great to see the authors extend their analysis in Figures 4 and 5 to region PM. This could be done with the existing data, simply extending the line of fixed vertical preference from PM > V1 > LM. If they do not see the same effect in PM, there should be some treatment of the PM/LM differences in the discussion.

We thank the reviewer for raising this important point. In the paper we show examples from area LM rather than PM because LM is larger and more distant from the focus, therefore offering us pixels that are remote in cortical space while being close to the focus in retinotopic space. PM, instead, lies close to the focus (the site of Picrotoxin application, and is in general hard to image retinotopically, especially in the horizontal dimension (it is harder to find the exact border between PM and V1). However, occasionally we were able to image PM and study propagation not only towards LM but also towards PM. For example, this is illustrated in the figure below (format as in Figure 5).

Supporting Figure 1: Seizure propagation recruits homotopic regions in LM and PM. (a) Map of retinotopy (preferred horizontal position) with a line starting from area PM, reaching to the focus in V1 and continuing to area LM, to join 50 regions of interest (ROIs) with same preferred vertical position as the focus in V1. (b) Single trial time course along the ROIs in panel a. Arrowheads mark the focus in V1 and secondary foci in PM and LM. Dashed line delineate the areal borders. (c) Seizure invasion delay for each ROI, averaged across seizures. Shaded area represents s.e.m.

Moreover, in the paper we go beyond LM in various ways:

- Fig 4: while we show and model the profile of activation only for V1 and LM, we show the maximum projection of interictal events across the entire field of view, where it is clear that activation extend also to higher visual areas. In particular, a strong lobe of activation is visible in area PM.
- Fig 5: While we show example timecourses for regions of interest in V1 and LM, the analysis of recruitment delay is extended to the whole field of view. This analysis shows that the higher visual areas, including PM, are all recruited earlier that would be predicted by distance alone
- Fig 6: The analysis of phase is performed pixelwise and shows that the map of delay is predicted by retinotopy in all higher visual areas, including PM.

2) I was somewhat confused by Figure 6. Is the point that locations that share retinotopy with the seizure initiation point have low latency delay? (If this is the case, showing seizure initiation points would help make this point clear - see minor point #4). Or alternatively, is the point that low latency delay is always mapped to horizontal retinotopy regardless of the initiation site? (This doesn't make much sense to me, but it is unclear from the figure). In any case, this analysis should be presented more clearly in the text/figure to avoid confusion.

We thank the reviewer for highlighting this section as needing work. We have now clarified that the key point is that locations that share preferred retinotopy with the seizure focus have low latency delay. As the reviewer suggested, we modified multiple figures to clearly label the position of the focus, and we superimposed the outlines of the visual areas to the delay map. Those suggestions were very useful and we thank the reviewer.

Minor concerns/suggestions:

1) Grammatical error in line 15 of abstract.

Fixed, thank you.

2) References 22/30: duplicate citation

Fixed, thank you.

3) Typo in Fig 1i: red triangles > blue triangles

Fixed, thank you.

4) It would be helpful to indicate the location of the electrode tip (seizure initiation point), particularly in Fig 3 and 6.

We revised all the figures to clearly indicate the electrode position in the relevant panels.

5) The methods section (and acknowledgements) refers to histology and confocal imaging, but I couldn't find any histology in the paper.

We did histology to confirm transgene expression in our mice, but we do not show any histology in the paper because all the details of the lines we used match the description from the Allen Institute/Jackson labs from which we obtained them. Therefore, we removed any mention of histology from Methods.

6) More details on the imaging system should be included in the "widefield calcium imaging" section of the methods. It wasn't clear to me whether the authors used a tandem lens or objective-based imaging system. The specs and part numbers on the major optics should also be included.

We have included a more detailed description of our imaging setup, which is indeed a microscope based on a tandem lens.

7) Two typos in lines 548 and 549: where > were.

Fixed, thank you.

Reviewers' Comments:

Reviewer #1 (Remarks to the Author):

The authors made major efforts to revise the manuscript. They addressed many concerns, clarified the scope and revised the text. The authors were up front about the limitations of extrapolating from their experimental model. In addition, they added some bits of data to demonstrate the degree to which the seizures they looked at generalized across the cortex and to compare variability in response between animals. More results were provided with regard to the behavioral measures reported that makes it clear that the mice were experiencing seizures as opposed to milder hyperactivity.

However, I still have a major concern that needs to be addressed. As I had mentioned in my previous review, the authors should not overstate that they studied epilepsy in this manuscript. The authors still claim they study focal epilepsy, but the model they use is not an epilepsy model in any way. It is an acute seizure model. For instance, they write on page 7 "Our characterization of the spatiotemporal flow of activity in focal epilepsy may provide a platform for further understanding the underlying mechanisms and for testing therapeutic approaches".

The major strength of the manuscript is technical. The main scientific novelty is limited to the description of the activity propagation which is highly consistent with the expectation from homotopic cortical connectivity.

Overall, I think the manuscript is significantly strengthened.

Reviewer #2 (Remarks to the Author):

Thanks for the changes, I think the paper has been quite improved. In particular, I am happy with the clarifications about the notion of functional connectivity, and clarified link with the functional specialization of V1, as well as better citations of the previous literature. I am also happy to see that the spiral wave dynamics are now much more explicitly mentioned and shown. I think this is a beautiful addition to the paper and makes it more coherent with preceding studies.

A question: would it be possible to post the code for analyzing the data? Even if the methods are not new to this paper, I think that would be a great resource to the other labs wanting to do such analyses.

Reviewer #3 (Remarks to the Author):

Rossi et al. have addressed my concerns and I support publication.

Response to reviewers

We thank the reviewers for their useful comments and criticisms. We reviewed the manuscript to meet their latest suggestions as detailed below.

Reviewer 1

The authors made major efforts to revise the manuscript. They addressed many concerns, clarified the scope and revised the text. The authors were up front about the limitations of extrapolating from their experimental model. In addition, they added some bits of data to demonstrate the degree to which the seizures they looked at generalized across the cortex and to compare variability in response between animals. More results were provided with regard to the behavioral measures reported that makes it clear that the mice were experiencing seizures as opposed to milder hyperactivity.

We are happy the reviewer appreciated our efforts to revise the manuscript and thank the reviewer for the positive comments.

However, I still have a major concern that needs to be addressed. As I had mentioned in my previous review, the authors should not overstate that they studied epilepsy in this manuscript. The authors still claim they study focal epilepsy, but the model they use is not an epilepsy model in any way. It is an acute seizure model. For instance, they write on page 7 “Our characterization of the spatiotemporal flow of activity in focal epilepsy may provide a platform for further understanding the underlying mechanisms and for testing therapeutic approaches”.

We do not wish to overstate the results of the paper. We have now changed the statement as follows: *‘Our characterization of the spatiotemporal flow of activity in focal cortical seizures may provide a platform for further understanding the underlying mechanisms and for testing therapeutic approaches.’*

The major strength of the manuscript is technical. The main scientific novelty is limited to the description of the activity propagation which is highly consistent with the expectation from homotopic cortical connectivity.

Overall, I think the manuscript is significantly strengthened.

Reviewer 2

Thanks for the changes, I think the paper has been quite improved. In particular, I am happy with the clarifications about the notion of functional connectivity, and clarified link with the functional specialization of V1, as well as better citations of the previous literature. I am also happy to see that the spiral wave dynamics are now much more explicitly mentioned and shown. I think this is a beautiful addition to the paper and makes it more coherent with preceding studies.

We thank the reviewer for these positive remarks about our revised manuscript.

A question: would it be possible to post the code for analyzing the data? Even if the methods are not new to this paper, I think that would be a great resource to the other labs wanting to do such analyses.

We added a ‘Data and code availability’ section to the paper. Here we provide the reader with a link to a repository where we will deposit an example dataset and the relevant code used for data analysis.

Reviewer 3

Rossi et al. have addressed my concerns and I support publication.

We are grateful for the reviewer’s appreciation of our work.